# Biogenic Silver Nanoparticles Processed Twice Using 8M Urea Exhibit Superior Antibacterial and Antifungal Activity to Commercial Chemically Synthesized Counterparts

**Terrence Ravine** [1,*] **, Qunying Yuan** [2] **and Makenna Howell** [1]

1   Department of Biomedical Sciences, University of South Alabama, Mobile, AL 36688, USA
2   Biological and Environmental Sciences, Alabama A&M University, Normal, AL 35762, USA
*   Correspondence: travine@southalabama.edu

**Abstract:** Biogenic silver nanoparticles (b-AgNPs) were produced extracellularly using a cell lysate of genetically modified *Escherichia coli* and subdivided into three groups. Each group received a different treatment to determine which one best removed residual cell lysate material. The first group was treated twice using only water (water ×2), the second using 8M urea once (8M urea ×1), and the third using 8M urea twice (8M urea ×2). Subsequently, each group was assessed for its ability to inhibit the growth of six bacterial and two fungal pathogens. Testing was accomplished using the minimum inhibitory concentration (MIC) method. Commercially produced c-AgNPs were included for comparison. In all cases, the b-AgNPs (8M urea ×2) demonstrated the greatest inhibition of microbe growth. Conversely, the commercial AgNPs failed to show any growth inhibition at 10 μg/mL the highest concentration tested. The greater antibacterial activity of the b-AgNPs (8M urea ×2) over both b-AgNPs (8M urea ×1) and b-AgNPs (water ×2) is thought to be due to a larger degree of biofunctionalization (coating) occurring during the two sequential 8M urea treatments.

**Keywords:** biogenic; silver nanoparticles; 8M urea; antimicrobial activity; minimum inhibitory concentration; functionalized

## 1. Introduction

Antimicrobial metallic nanoparticles (NPs) are routinely incorporated into a variety of materials to prevent microbes such as bacteria and fungi from destroying them. For example, adding biocidal NPs to textiles can stop/slow down material deterioration, thereby extending its use. Silver nanoparticles (AgNPs) are very effective at either reducing the number of contaminating microbes or completely eliminating them. Consequently, this helps to limit their spread. Biocidal AgNPs have been integrated into wound dressings and medical devices such as dental implants, cardiovascular implants and imaging probes. Additional medical applications include AgNPs use as antibiofilm agents, antitumor agents, and bone healing promoters. Moreover, AgNPs are layered onto optoelectronic devices used in the electronic industry to control microbial growth [1,2].

The usefulness of AgNPs has led to an explosive growth of the nanobiotechnology industry. This caused a corresponding increase in AgNPs production. In 2011, it was estimated that nearly 280 tons of AgNPs were produced for commercial or industrial use [3]. That number increased to about 500 tons of annual global production in 2021 [4]. This figure also includes AgNPs produced for use in the electronics industry. The silver nanoparticles market was valued at USD 1.5 billion in 2020, and it is anticipated that the market will reach USD 6.6 billion by 2030 [5].

Chemical synthesis represents a major method for rapidly producing large numbers of AgNPs. This process involves using a strong reducing agent to convert a chemical compound such as silver nitrate ($AgNO_3$) into a metallic particle measuring in the nanometer range. The use of silver-based nanomaterials was initially hindered due to their instability

from oxidation. To remedy this problem, a stabilizer is added to keep the AgNPs from undergoing rapid oxidation [2]. Although chemical synthesis is very efficient, it also produces a large volume of harsh chemical waste that eventually makes its way into the environment. Accordingly, alternative green synthesis methods are being employed to help reduce an ever-increasing volume of chemical wastes associated with silver nanomaterial production.

Green synthesis can occur by either chemical means or biogenesis. Both processes produce AgNPs that have less of an environmental impact [6]. Chemical green synthesis involves the use of natural substances, such as plant extracts, to produce chemically synthesized AgNPs (c-AgNPs) but with less associated toxic waste. As the name suggests, biogenic AgNPs (b-AgNPs) are produced by either giving a silver substrate to living microbe-like bacteria, using microbe culture supernatants, or using microbe whole-cell lysates [7]. b-AgNPs produced by this method appear to hold promise in combating biofilm-associated infections, since biofilms tend to harbor antibiotic-resistant pathogens [8].

AgNPs effectively control the growth of a wide range of microbes, including antibiotic-resistant bacteria causing infections. Still, their use is hindered by an intrinsic level of toxicity to humans and associated environmental hazards [9]. Although the demand for AgNPs keeps growing, there has not been a corresponding effort to reduce their impact on the environment, regardless of their synthesis method. Likewise, other metallic nanoparticles with equivalent or greater antimicrobial activity to AgNPs but with reduced toxicity are being sought as a potential replacement. However, the demonstrated antimicrobial effectiveness of AgNPs makes it a prime contender for continued use until a suitable replacement is found. Consequently, producing long-lasting b-AgNPs exhibiting broad-spectrum antimicrobial action at a lower working concentration is highly desired.

The novelty of this study involves two separate findings. First, the differences in antimicrobial effectiveness seen between tested b-AgNPs and c-AgNPs demonstrates that b-AgNp are superior to c-AgNPs in controlling microbe growth. Second, the differences seen between the antimicrobial effectiveness of each b-AgNP type appears to be a product of variation in their post-production processing. We now report on b-AgNPs exhibiting excellent antimicrobial activity at substantially lower concentrations than commercial c-AgNPs.

## 2. Materials and Methods

### 2.1. Nanoparticle Preparation

Cell lysates from previously characterized recombinant *Escherichia coli* DH5$\alpha$ cells were used to produce b-AgNPs [10,11]. In doing so, a 10 µL aliquot of 15% glycerol stocks of *Escherichia coli* DH5$\alpha$ cells was transferred into 10 mL of LB medium containing 100 µg/mL carbenicillin and grown overnight at 37 °C on a shaker incubator at 250 rpm. The overnight cell culture was then centrifuged, and cells were resuspended in 100 mL of LB in the presence of carbenicillin and re-incubated at 37 °C while periodically monitoring optical density (OD).

When OD600 reached 0.9, cells were centrifuged at 4500 G for 15 min. The resulting cell pellet was washed with 25 mL of 50 mM sodium phosphate buffer (pH 9) and re-suspended in 2 mL of 50 mM sodium phosphate buffer (pH 9). Subsequently, cells were sonicated on ice for three cycles of 60 pulses at an output of 20%, with a 1 min interval between each cycle. The lysate was spun down to remove the cell debris. The soluble intracellular extract was added to 100 mL of 50 mM sodium phosphate buffer (pH 9) containing 1 mM AgNO$_3$. This cell-free reaction mix was incubated at 37 °C with continuous shaking at 250 rpm for 4 days. The b-AgNPs were then collected by centrifugation at 5000 rpm for 15 min. The pellets were sonicated in 50 mM sodium phosphate buffer (pH 9) for 1 min while on ice and centrifuged at 3000 G for 4 min.

The recovered b-AgNPs were separated into 3 groups, each one being processed somewhat differently. The first group of pelleted b-AgNPs were re-suspended in pure H$_2$O by sonication on ice for 1 min. The b-AgNPs were collected by centrifugation at 17,000 rpm for 15 min at 4 °C. This wash step was repeated once and then the b-AgNPs were resuspended in pure H$_2$O. This set was designated as b-AgNPs (water ×2). The

second group of b-AgNPs were sonicated in 8M urea on ice for 1 min, placed on an orbital shaker at 360 rpm for 30 min at 37 °C, and then pelleted by centrifugation at 17,000× $g$ for 20 min. The pellet was washed twice with pure $H_2O$ and resuspended by sonication in pure $H_2O$. This set was designated as b-AgNPs (8M urea ×1). The third group was processed similarly to the b-AgNPs (8M urea ×1) but included a second sonication in 8M urea of the supernatant recovered from the first 8M urea wash. The b-AgNPs were washed with pure $H_2O$ twice and resuspended in pure $H_2O$. This set was designated as b-AgNPs (8M urea ×2).

Commercial c-AgNPs were obtained from both Alfa Aesar (Cat # J67099) through Fisher Scientific (Waltham, MA, USA) and SkySpring Nanomaterial s(Cat # 0115CY); (Houston, TX, USA). The Alfa Aesar c-AgNPs were 100 nm in size and adjusted to 0.02 mg/mL in 2 mM sodium citrate. The SkySpring Nanomaterials were also 100 nm size but were received in a powdered form. The b-AgNPs (water ×2), b-AgNPs (8M urea ×1), and b-AgNPs (8M urea ×2) samples were all received suspended in pure water. They were subsequently freeze-dried using a Labconco lyophilizing unit (Kansa City, MO, USA). Then, both the SkySpring c-AgNPs powder and lyophilized b-AgNPs were adjusted to 0.02 mg/mL concentration in 2 mM sodium citrate to match the Alfa Aesar c-AgNPs. Two-fold serial dilutions of each AgNPs type were prepared in 1.5 mL amber conical-bottom microcentrifuge tubes (VWR, Atlanta, GA, USA) using sterile deionized water. All dilutions were stored at 4 °C until tested.

### 2.2. Tested Microbes

The studied bacteria included *Escherichia coli* (ATCC 25922), *Staphylococcus aureus* (ATCC 29213), and *Pseudomonas aeruginosa* (ATCC 27853). Antibiotic-resistant strains (ARS) included methicillin-resistant *Staphylococcus aureus* or MRSA (ATCC 45300), vancomycin-resistant *Enterococcus faecalis* or VRE (ATCC 51299), and extended spectrum beta-lactamase *Escherichia coli* or ESBL (Patient isolate). Tested fungi included *Candida albicans* (ATCC 60193) and *Aspergillus fumigatus* (KM 8001); (Table 1). Stock cultures were maintained by periodic passage on growth supportive nutrient agar and incubated at 37 °C under ambient conditions (no $CO_2$). A single well-isolated colony of a test bacterium was transferred to a fresh agar plate and incubated 24 h before each assay. Fungi were incubated for 48 h before antimicrobial testing.

**Table 1.** Characteristics of Tested Bacteria and Fungi (n = 8).

| Features | Microbes | | | | | |
|---|---|---|---|---|---|---|
| | *Enterococcus faecalis* [1] | *Escherichia coli* [2] | *Pseudomonas aeruginosa* | *Staphylococcus aureus* [3] | *Candida albicans* | *Aspergillus fumigatus* |
| Microbe type | Bacterium | Bacterium | Bacterium | Bacterium | Fungus | Fungus |
| Gram stain reaction | Positive | Negative | Negative | Positive | N/A | N/A |
| Morphology | Spherical (coccus) | Rod-shaped (bacillus) | Rod-shaped (bacillus) | Spherical (coccus) | Yeast (oval) | Mold (filamentous) |
| Metabolism | Facultative anaerobic | Facultative anaerobic | Obligate aerobic | Facultative anaerobic | Fermentation | Nutrient assimilation |

[1] *Enterococcus faecalis* was a vancomycin-resistant enterococcus (VRE) strain. [2] Included both an extended spectrum beta-lactamase (ESBL) *Escherichia coli* and non-ESBL *Escherichia coli* strains. [3] Included both a methicillin-sensitive (MSSA) and methicillin-resistant (MRSA) *Staphylococcus aureus*.

For all tested bacteria, 2–3 well isolated colonies were transferred to 3 mL cation-adjusted Muller-Hinton (M-H) broth (Remel, Lenexa, KS, USA) in a sterile tube with a glass screw top. The suspension was adjusted to match a 0.5% McFarland turbidity standard using a DEN-1 McFarland Densitometer (Grant-Bio, Beaver Falls, PA, USA), which results

in ~$1.0 \times 10^8$ colony-forming units (CFU)/mL$^{-1}$. An additional 1:100 dilution of the 0.5% adjusted sample was made in M-H broth to yield approximately $1.0 \times 10^6$ colony-forming units (CFU)/mL$^{-1}$.

For *Candida albicans* yeast, 4–6 well isolated colonies were removed from a 48-h nutrient agar plate and placed into a sterile glass screw top tube containing 3 mL Muller-Hinton (M-H) broth (Remel, Lenexa, KS, USA). The suspension was adjusted to match a 0.5% McFarland turbidity standard, as was previously described for bacteria. An additional 1:100 dilution was not performed.

For *Aspergillus fumigatus* mold, five 48 h nutrient agar plates containing actively growing mold colonies were flooded with 5 mL of 0.9% sterile saline. Intact fungal colonies were disrupted using a sterile cell spreader (Fisher Scientific 14-665-231, Pittsburgh, PA, USA) by moving the spreader vigorously back and forth over the entire plate surface creating a slurry. This suspension was transferred to a 50 mL conical centrifuge tube (Fisher Scientific12-565-271, Pittsburgh, PA, USA) and vigorously vortexed for 30 s on the highest mixer setting. The resulting solution was adjusted to match a 0.5% McFarland turbidity standard.

The number of either bacteria or fungi in MIC starting inoculum was determined as follows. Ten-fold serial dilutions were prepared of the turbidity adjusted inoculum in 0.9% sterile saline. A 100 µL (0.1 mL) aliquot of each dilution was transferred to a separate nutrient agar plate and spread over its surface using a sterile cell spreader. Plates were incubated for either 24 or 48 h, inspected for growth, and colonies counted. The colony forming units (CFU)/mL$^{-1}$ was determined by counting the number of bacterial or fungal colonies present on a dilution plate demonstrating between 10 and 100 colonies. This number was multiplied by the reciprocal of the serial dilution and then multiplied by ten to account for the 100 µL (0.1 mL) sample volume distributed to each count plate.

### 2.3. Minimum Inhibitory Concentration (MIC) Testing

A 100 µL sample of each test nanoparticle dilution was distributed to separate wells of a 96-well microtiter plate (Figure 1). Each well containing a nanoparticle dilution received a 100 µL aliquot of the microbe inoculum being tested. This resulted in a 1:2 dilution of each nanoparticle concentration in test wells, ranging from 10 µg/mL down 0.04 µg/mL. A negative growth control consisting of 200 µL of the M-H broth, without microbe, was included to detect potential exogenous broth contamination. A separate positive growth control consisting of 100 µL of only 2 mM sodium citrate (no nanoparticles) plus 100 µL aliquot of the microbe inoculum was used to see if 2 mM sodium citrate alone contributed to growth inhibition. The inoculated microtiter plate was covered with a fitted lid and sides wrapped with Parafilm® to prevent sample desiccation during incubation. The plate was placed inside a MaxQ 4450 shaking incubator (ThermoFisher Scientific, Waltham, MA, USA) and held for either 24 h (bacteria) or 48 h (fungi) at 37 °C under ambient conditions with constant rotation at 100 RPM.

PMI 1640 medium supplemented with 2% glucose is typically used for fungal MIC testing in a clinical lab setting [12,13]. Instead, the current study used M-H broth to perform fungal MIC testing. This allowed for a more direct comparison to the bacterial MIC results. M-H broth supports fungal growth and has been used to perform *Candida albicans* germ tube testing in lieu of serum [14]. Additionally, a control well containing either *Candida albicans* or *Aspergillus fumigatus* test inoculum without nanoparticles demonstrated appreciable growth in each MIC accomplished.

In addition to initial MIC testing, all 3 b-AgNPs and Alfa Aesar c-AgNPs were retested a year later using the same methicillin-sensitive *Staphylococcus aureus* (MSSA) bacterial strain. This was done to determine the amount of residual antimicrobial activity, if any, that was present after long-term storage at 4 °C. The SkySpring c-AgNPs, which had not been acquired prior to initial MIC testing, were also included in the stability assay.

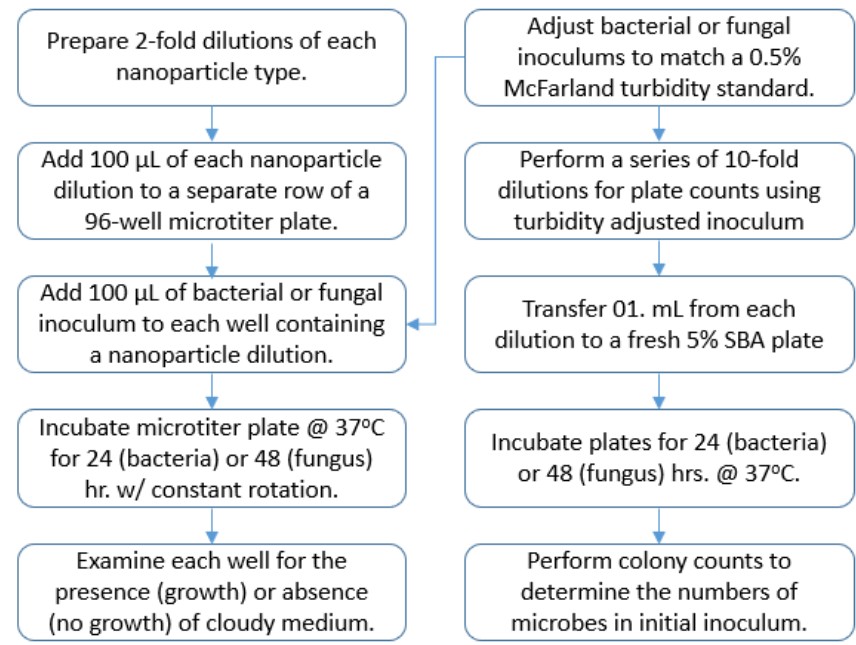

**Figure 1.** Testing schematic. Simplified diagram outlining general steps accomplished during the testing process.

*2.4. Spectroscopy*

Direct sampling of each 0.02 mg/mL in 2 mM sodium citrate AgNPs solution was performed for both visible (vis-) and Fourier-transform infrared (FT-IR) spectroscopy. Vis- spectroscopy was accomplished using a Thermo Scientific Evolution 350 UV-VIS spectrometer and Thermo Scientific "Insight 2" software being used to display the resulting data (Waltham, MA, USA). FT-IR was accomplished using a Thermo Scientific Nicolet iS50 FTIR with built-in ATR accessory (Waltham, MA, USA). The AgNPs for X-ray energy-dispersive spectroscopy (EDS) analysis were resuspended in pure water by sonication prior to analysis. A Hitachi HD2700 aberration-corrected scanning transmission electron microscope (Hitachi, Tokyo, Japan) was used to acquire the EDS spectrum.

*2.5. TEM*

The 0.02 mg/mL in 2 mM sodium citrate AgNPs samples were prepared for transmission electron microscopy (TEM) as follows. Approximately 5.0 mL of each sample was centrifuged using an Eppendorf centrifuge 5810R (Enfield, CT, USA) at $3000\times g$ for 5 min and large precipitates were discarded. Samples were sonicated at preset pulse cycle $5\times$ using a Fisher Scientific FB120 sonicator equipped with a Qsonica model CL-18 probe (Newtown, CT, USA). A 2 mL aliquot of supernatant was then transferred to a 2.0 microcentrifuge tube and centrifuged using a Hermle Labnet Z 323 K (Gosheim, Germany) set to $17,000\times g$ for 30 min. A 1.5 mL aliquot of supernatant was removed and discarded. The remaining 0.5 mL was vigorously vortexed and pipetted up and down several times to thoroughly mix the nanoparticles. The resulting suspension was sonicated again at the same setting. A 2 μL sample of well-suspended nanoparticles was separately applied to a pioloform-filmed 400 mesh Cu grids (Electron Microscopy Sciences). Grids were allowed to air dry completely before being inspected. Subsequent TEM examination was accomplished using a JEOL JEM 1400 operating at 80 kV.

*2.6. Particle Sizing*

The average size and distribution range of AgNPs were measured using a Model ZEN 1600 Zetasizer (Malvern, UK). This instrument uses dynamic light scattering (DLS) to determine particle characteristics. The Zetasizer unit was calibrated prior to particle analysis by using Nanosphere (61 nm ± 4 nm), an NIST traceable latex standard (ThermoFisher,

Waltham, MA, USA). The duration of the count ranged from 70–290 s. The mean value of peak #1 representing the maximum intensity was used to report the particle diameter (nm) and width (nm). Zeta potential measurements were not supported by this analyzer model.

Direct measurements of both b-AgNPs (8M urea ×2) and Alfa Aesar c-AgNPs were also accomplished from TEM microphotographs using ImageJ software. Twenty random measurements were taken for each AgNPs sample type.

## 3. Results

### 3.1. MIC Testing

The b-AgNPs (8M urea ×2) demonstrated the greatest antimicrobial activity in 8/8 (100%) microbes by MIC testing (Tables 2–4). The lowest b-AgNPs (8M urea ×2) MIC of 0.31 μg/mL were recorded for the bacteria *Escherichia coli*, *Pseudomonas aeruginosa*, and ESBL- *Escherichia coli* and the highest MIC of 2.50 μg/mL for the mold *Aspergillus fumigatus*. With the exception of *Aspergillus fumigatus* mold, all 3 b-AgNPs types were effective against each tested bacterial and fungal strain. Listed in order of effectiveness against *Candida albicans* yeast were b-AgNPs (8M urea ×2) at 1.25 μg/mL, b-AgNPs (8M urea ×1) at 2.50 μg/mL, and b-AgNPs (water ×2) at 5.00 μg/mL. In contrast, no antimicrobial activity was exhibited by the Alfa Aesar c-AgNPs at 10 μg/mL, the highest concentration evaluated, in 8/8 (100%) of tested microbes. A similar finding of >10 μg/mL was noted for SkySpring c-AgNPs (SkySpring) included in fungal and stability testing. By itself, the b-AgNPs (8M urea ×2) MIC range of 0.31–2.50 μg/mL represents an effective concentration at least 4-32X greater than c-AgNPs (>10 μg/mL).

**Table 2.** Minimum inhibitory concentrations (MIC) of tested nanoparticles against *Escherichia coli*, *Pseudomonas aeruginosa*, and *Staphylococcus aureus* after 24 h incubation at 37 °C in ambient air.

| Nanoparticle | MIC (μg/mL) | | |
| --- | --- | --- | --- |
| | *Escherichia coli* ($5.0 \times 10^5$) | *Pseudomonas aeruginosa* ($7.5 \times 10^5$) | *Staphylococcus aureus* (MSSA) ($7.5 \times 10^5$) |
| b-AgNPs (water ×2) | 1.25 | 1.25 | 2.50 |
| b-AgNPs (8M urea ×1) | 1.25 | 1.25 | 1.25 |
| b-AgNPs (8M urea ×2) | 0.31 | 0.31 | 0.62 |
| c-AgNPs (Alfa Aesar) | ≥10.0 | ≥10.0 | ≥10.0 |

MSSA = Methicillin-sensitive *Staphylococcus aureus*, ( ) = Colony-forming units per mL$^{-1}$.

**Table 3.** Minimum inhibitory concentrations (MIC) of tested nanoparticles against antibiotic resistant strains of *Enterococcus faecalis*, *Escherichia coli*, and *Staphylococcus aureus* after 24 h incubation at 37 °C in ambient air.

| Nanoparticle | MIC (μg/mL) | | |
| --- | --- | --- | --- |
| | *Escherichia faecalis* (VRE) ($0.6 \times 10^5$) | *Escherichia coli* (ESBL) ($5.0 \times 10^5$) | *Staphylococcus aureus* (MRSA) ($5.5 \times 10^5$) |
| b-AgNPs (water ×2) | 2.50 | 1.25 | 2.50 |
| b-AgNPs (8M urea ×1) | 2.50 | 0.62 | 2.50 |
| b-AgNPs (8M urea ×2) | 1.25 | 0.31 | 0.62 |
| c-AgNPs (Alfa Aesar) | ≥10.0 | ≥10.0 | ≥10.0 |

VRE = Vancomycin-resistant enterococcus, ESBL = Extended spectrum beta-lactamase, MRSA = Methicillin-resistant *Staphylococcus aureus*, ( ) = Colony-forming units per mL$^{-1}$.

**Table 4.** Minimum inhibitory concentrations (MIC) of tested nanoparticles against *Candida albicans* yeast & *Aspergillus fumigatus* mold after 48 h incubation at 37 °C in ambient air.

| | MIC (µg/mL) | |
| --- | --- | --- |
| **Nanoparticle** | *Candida albicans* $(1.0 \times 10^5)$ | *Aspergillus fumigatus* $(1.0 \times 10^5)$ |
| b-AgNPs (water ×2) | 5.00 | ≥10 |
| b-AgNPs (8M urea ×1) | 2.50 | ≥10 |
| b-AgNPs (8M urea ×2) | 1.25 | 2.50 |
| c-AgNPs (Alfa Aesar) | ≥10 | ≥10 |
| c-AgNPs (SkySpring) | ≥10 | ≥10 |

( ) = Colony-forming units per $mL^{-1}$.

### 3.2. Spectroscopy

Vis-spectroscopy of commercially produced Alfa Aesar c-AgNPs revealed the presence of two absorbance peaks. The first peak, seen at 418.6 nm, was very near 420 nm, a value consistent with silver nanoparticles (Figure 2) [6]. A second broader peak was noted at 525.5 nm. The b-AgNPs (water ×2) showed a peak at 406.6 nm, near the first Alfa Aesar c-AgNPs peak, but a similar peak was not seen for either b-AgNPs (8M urea ×1) or b-AgNPs (8M urea ×2). All three b-AgNP types demonstrated peaks clustered around 570 nm ranging from 562.8 nm to 584.3 nm. See Supplemental Data for individual spectra and Voight residual fit (Available online: https://www.mathworks.com/matlabcentral/fileexchange/57603-voigt-line-shape-fit (accessed on 25 October 2022)).

**Figure 2.** Visible Spectra of Tested Nanoparticles. Absorbance peaks are shown for all three b-AgNPs (A–C), Selenium NPs unrelated to current study (D) and Alfa Aesar c-AgNPs (E).

FT-IR analysis of the b-AgNPs (8M urea ×2) and Alfa Aesar c-AgNPs samples demonstrated mostly similar absorbance patterns, with the exception of two small peaks seen for Alfa Aesar c-AgNPs at 2359.87 $cm^{-1}$ and 2336.46 $cm^{-1}$ wavelengths (Figure 3). Both peaks occurred in the mid-IR wavelength region. Two further differences noted were peaks at 667.86 $cm^{-1}$ for Alfa Aesar c-AgNPs and 507.55 $cm^{-1}$ for b-AgNPs (8M urea ×2). Neither appear to be in a frequency range associated with a characteristic functional group.

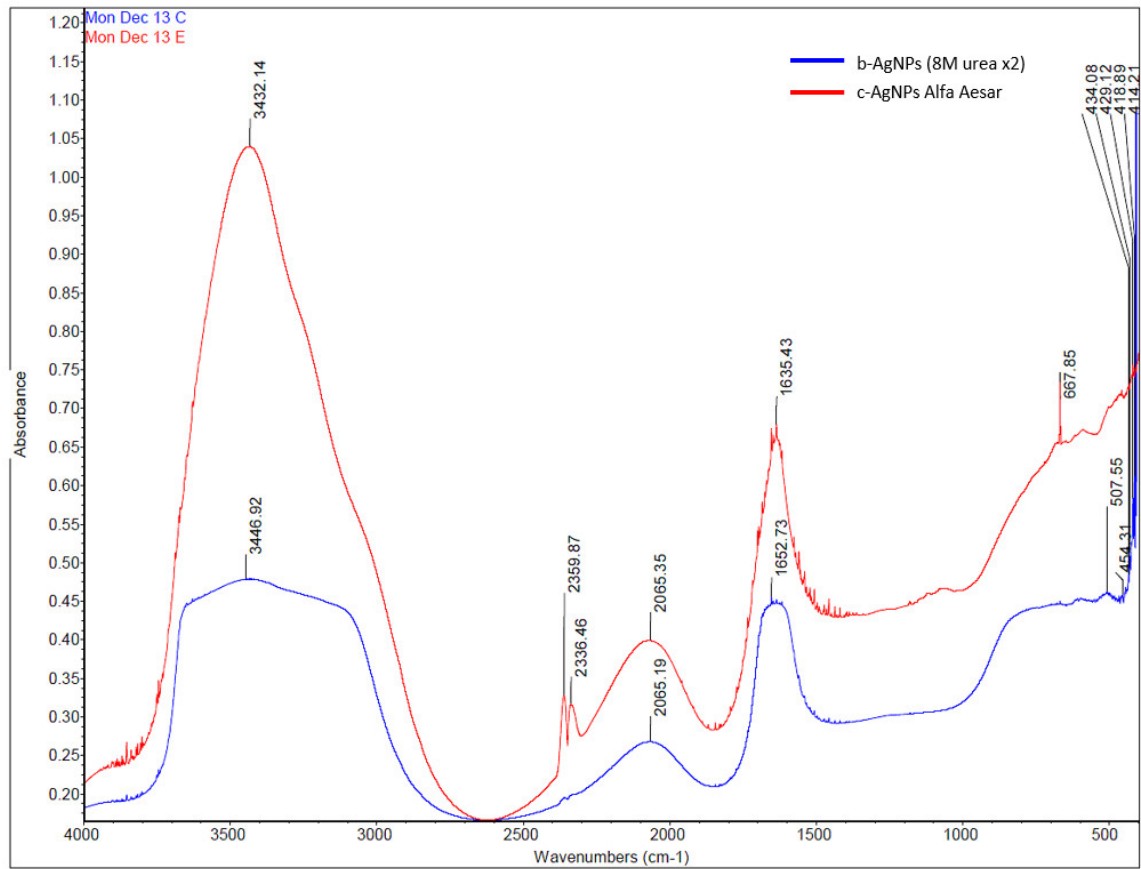

**Figure 3.** Fourier-transform infrared spectroscopy (FT-IR). Absorbance patterns of (C/blue) b-AgNPs (2× 8M urea) and (E/red) Alfa Aesar c-AgNPs.

The EDS spectrum revealed the presence of an elemental Ag signal starting near 3 kEv (Figure 4). However, more prominent was a C peak of greater intensity seen below 1 kEv. Additional peaks of varying intensity, but lower than that of either C or Ag, were observed for O, N, and P (listed in decreasing order) along with much smaller peaks for both S and CL.

### 3.3. TEM

A direct comparison of both AgNPs (8M urea ×2) and Alfa Aesar c-AgNPs TEM microphotographs showed mostly sphere-shaped AgNPs, but also some differences between them. The AgNPs (8M urea ×2) image not only revealed numerous single NPs, when viewed at 12,000×, but also the presence of an occasional small aggregate (Figure 5). In contrast, when viewed at 3000×, the c-AgNPs were more consistent in size without similar aggregates. Examination of the b-AgNPs (water ×2) and b-AgNPs (8M urea ×1) photomicrographs (not shown) also showed spherical-shaped AgNPs along with some small aggregates.

### 3.4. Size Determination

Zetasizer analysis revealed that the Alfa Aesar c-AgNPs were the most consistent in size measuring ~129 nm (see Table 5). This value was close to the advertised core size of 100 nm with a range from 98–115 nm indicated on the accompanying certificate of analysis. On the contrary, greater heterogeneity was seen between the sizes of the 3 b-AgNPs. Listed in increasing size order was the b-AgNPs (8M urea ×2) at ~156 nm, b-AgNPs (8M water ×2) at ~176 nm, and b-AgNPs (8M urea ×1) at ~329 nm. It should also be noted that clumps were visible in the b-AgNPs (8M urea ×1) sample, which may have contributed to its a larger size measurement. These clumps could not be easily dispersed by sonication. It

should also be mentioned that a sample from a more recent batch of b-AgNPs (8M urea ×2) was sent for size analysis. Peak 1 involved 147.4 nm particles with a Z-average of 140.9 nm and a zeta potential of −44.4 mV. The zeta potential indicated good b-AgNPs (8M urea ×2) stability. Measurement was performed by DLS using a Malvern Zetasizer Nano ZS while b-AgNPs (8M urea ×2) were suspended in pure water.

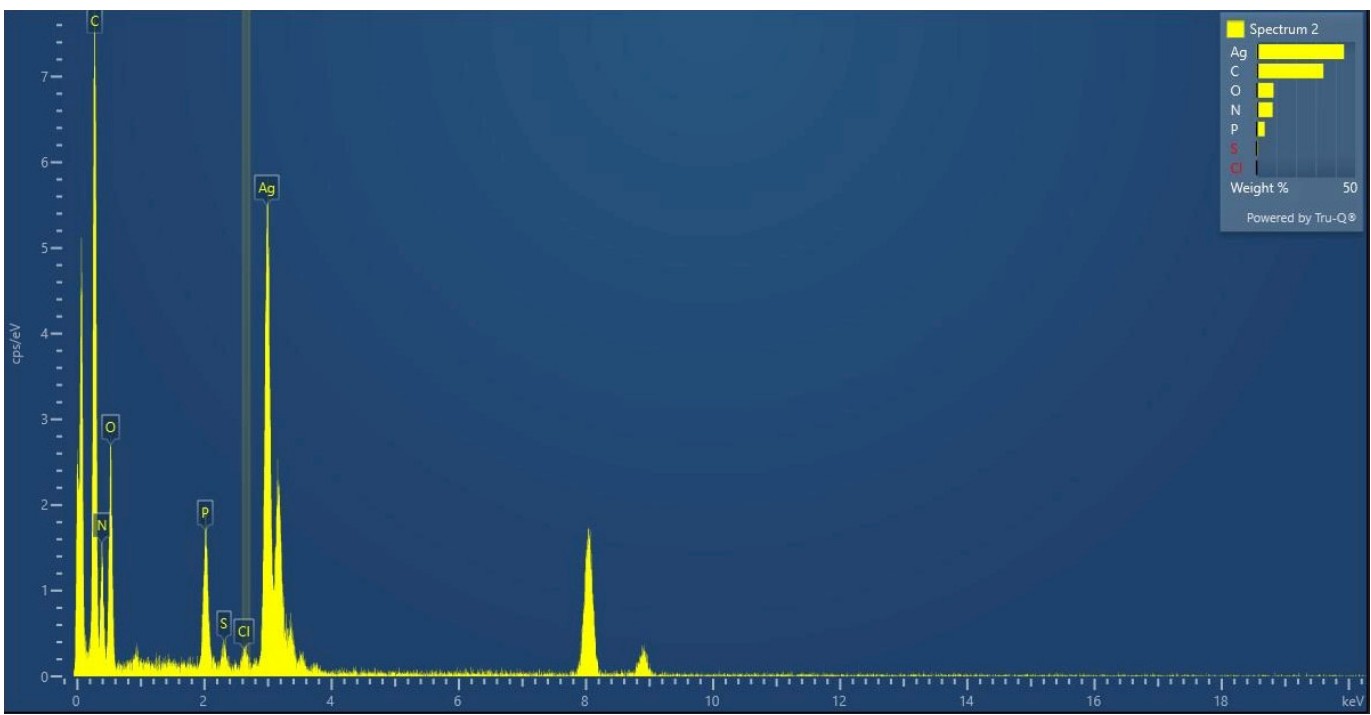

**Figure 4.** Energy dispersive spectroscopy (EDS). X-ray spectrum of b-AgNP (8M urea ×2) indicating Ag, C, O, N, P along with small amounts of P, CL.

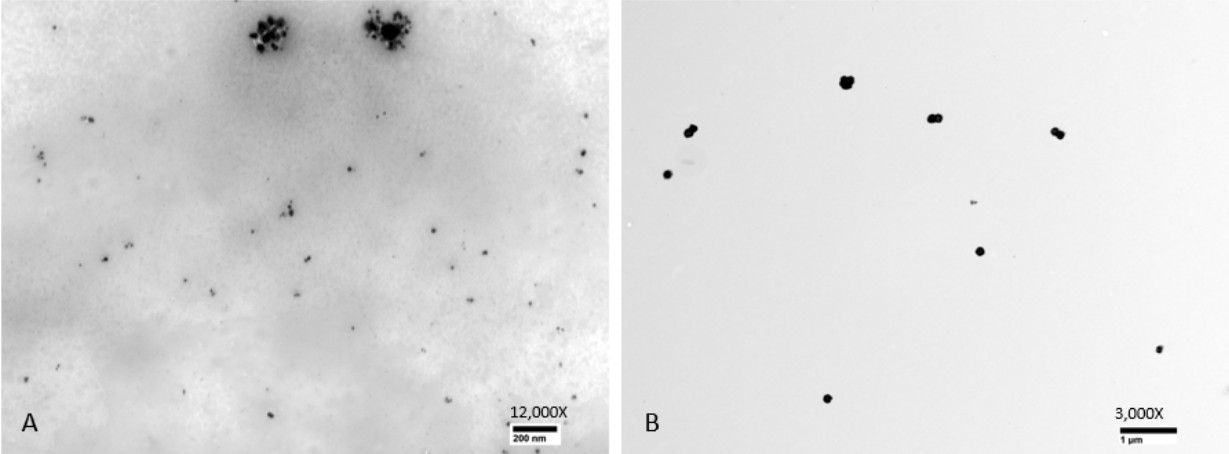

**Figure 5.** Transmission electron micrographs. Nanoparticle preparations of (**A**) b-AgNPs (8M urea ×2) at 12,000× magnification and (**B**) Alfa Aesar c-AgNPs at 3000× magnification.

**Table 5.** Particle measurements by DLS method using a Malvern ZEN 1600 Zetasizer.

| Nanoparticle | Size | Width |
|---|---|---|
| AgNP (water ×2) | 176.3 nm | 12.95 nm |
| AgNP (8M urea ×1) | 329.8 nm | 56.66 nm |
| AgNP (8M urea ×2) | 156.1 nm | 12.98 nm |
| AgNP (commercial) | 129.2 nm | 40.55 nm |

ThermoFisher latex particle control (61 nm ± 4) = 65.2 nm.

TEM photomicrograph analysis using ImageJ software revealed a b-AgNPs (8M urea ×2) average diameter of 28.7 nm ranging in size from 15.5–49.1 nm. Measurements of b-AgNPs (8M urea ×2) aggregates ranged from 148–170 nm. In contrast, the Alfa Aesar c-AgNPs were determined to be 143.5 nm ranging from 103.3–179.91 nm. Aggregates were not noted in the commercial sample.

## 4. Discussion

Collectively, these findings indicate that the b-AgNPs produced using cell lysate material are indeed silver nanoparticles. Support is offered by Vis-spectrum, FT-IR, TEM, and EDS results. Furthermore, findings suggest that processing b-AgNPs twice increases their antimicrobial activity against microorganisms.

The vis-spectrum revealed two items of interest relating to the Alfa Aesar c-AgNPs (Figure 2). First, the peak intensities at both 418.6 nm and 525.5 nm were markedly higher than any seen for b-AgNPs. This appears to indicate a relative difference in concentrations between the b-AgNPs and Alfa Aesar c-AgNPs. This finding correlates well with similar differences in peak amplitude between these two AgNPs demonstrated by FT-IR analysis. It lends support to the b-AgNPs, especially the b-AgNPs (8M urea ×2), being more bioactive than the Alfa Aesar c-AgNPs when used at a lower starting concentration. Second, there was an unexpected peak at 525.5 nm. An examination of the accompanying Alpha Aesar c-AgNP safety data sheet (SDS) listed water, sodium citrate (dihydrate), and silver as components [15]. The significance of this peak remains unknown despite an inquiry sent to Alfa Aesar.

Another interesting finding was a peak at 658.0 nm for the b-AgNPs (8M urea ×2) that was absent in either b-AgNPs (water ×2) or b-AgNPs (8M urea ×1). Generally speaking, absorption by AgNPs depends on the particle size, particle shape, dielectric medium, and chemical surroundings [16]. Ejbarah reported AgNP absorption peaks between 420 and 480 nm. These same peaks were seen to shift to a longer wavelength as particle size increased [17]. Barbar et al. reported 400 nm absorption peaks for c-AgNPs made using different concentrations of $AgNO_3$ and 0.5 mM trisodium and 0.3 mM sodium borohydride stabilizers [18]. In the current study, each of the tested AgNPs was suspended in 2 mM sodium citrate, which has a λ max near 210 nm [19]. This lower wavelength tends to eliminate any possible contribution of the 2 mM sodium citrate stabilizer to the noted peaks.

The presence of multiple absorbance bands detected by FT-IR for b-AgNP (8M urea ×2) and Alfa Aesar c-AgNP suggests that both have functional groups affixed to their surfaces (Figure 3). For example, the broad absorbance band seen at 3446.92 cm$^{-1}$ for b-AgNP (8M urea) represents either a single peak N-H stretch, a hydrogen-bonded O-H stretch, or possibly both. As noted for vis-spectrum, the heights of the Alfa Aesar c-AGNP peaks recorded by FT-IR were noticeably higher than the peaks for b-AgNPs (8M urea ×2).

The detection of C, O, N, P and small amounts of Cl and S by EDS analysis suggests the presence of organic matter (Figure 4). A minor amount of P was detected, which advocates that the silver nitrate did not react with the phosphate buffer to precipitate as salt but instead formed b-AgNPs. The organic matter may be residual cell lysate material still present after b-AgNP processing. It is more likely that some organic molecules in cell lysate became associated with AgNP surfaces while in contact with each other. The addition of 8M urea during processing may have further promoted the coating or functionalization of

the b-AgNPs with free cell lysate molecules. An appropriate functional group expressed on AgNP surfaces is all that is required for binding available biomolecules [1,9].

Examination of TEM photomicrographs of both b-AgNPs (8M urea ×2) and Alfa Aesar c-AgNPs revealed the presence of small particles in the expected nanoparticle range. The only noticeable difference between the two groups being the small aggregates seen in the b-AgNPs (8M urea ×2) photomicrographs that were absent in those for the Alfa Aesar c-AgNPs. The aggregates did not appear to have any negative impact on b-AgNP (8M urea ×2) antimicrobial activity. By direct measurement, the b-AgNPs (8M urea ×2) were noticeably smaller than those detected by Zetasizer measurements: 28.7 nm versus 156.1 nm, respectively (Figure 6). Measurements of aggregates ranged from 148–170 nm. This suggests that the aggregates contributed more to the 156.1 nm b-AgNPs (8M urea ×2) Zetasizer measurements than did individual particles. This observation underlines the value of evaluating DLS results by performing a direct measurement of AgNPs using SEM or TEM photomicrographs to detect aggregates. This is especially true when using earlier, less sensitive DLS instrumentation.

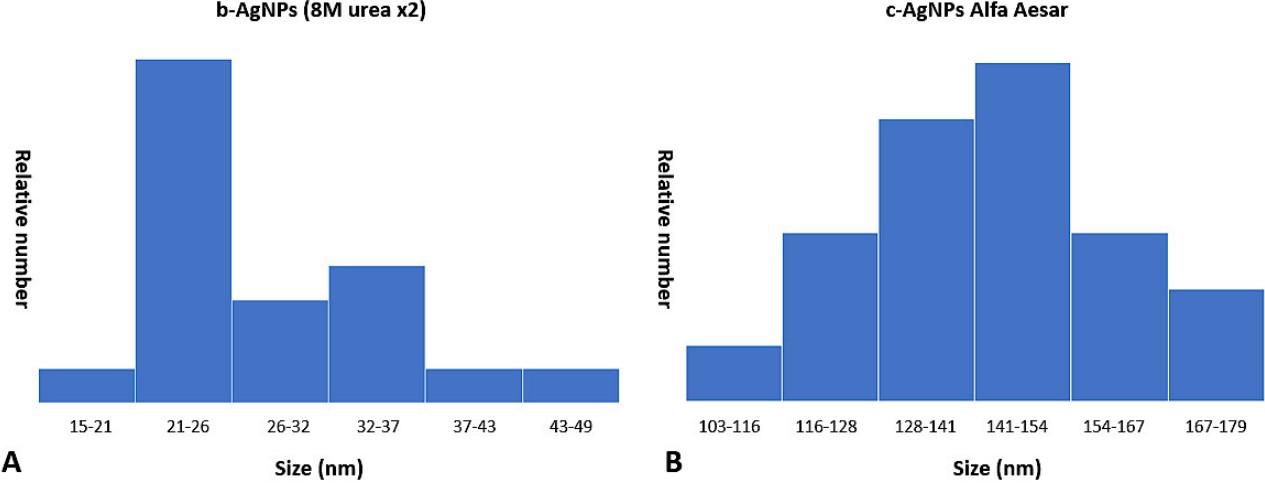

**Figure 6.** Histograms. Panel (**A**) shows the size distribution of b-AgNPs treated twice with 8M urea while panel (**B**) shows Alfa Aesar c-AgNPs (TEM measurements).

Concerning bioactivity, all three b-AgNPs types were effective against 6/6 (100%) bacteria by MIC testing (Tables 3–5). In all assays, the b-AgNPs (8m urea ×2) were the most effective at controlling bacterial growth. Both the b-AgNPs (8m urea ×1) and b-AgNPs (water ×2) showed mostly comparable values. Here, b-AgNPs (8M urea ×1) and b-AgNPs (water ×2) were equal in effectiveness in 4/6 (~67%) tested bacteria. The two exceptions to this equal effectiveness were *Staphylococcus aureus* (MSSA) and ESBL *Escherichia coli*. Here, the b-AgNPs (8M urea ×1) were seen to be more effective than b-AgNPs (water ×2). Similarly, all three b-AgNPs also showed good growth control of *Candida albicans* yeast. The b-AgNPs (8M urea ×2) showed the best control with an MIC of 1.25 μg/mL followed by b-AgNPs (8M urea ×1) with an MIC of 2.50 μg/mL, and b-AgNPs (water ×2) with an MIC of 5.0 μg/mL.

An MIC concentration of 5.0 μg/mL was required for the b-AgNPs (8M urea ×2) to control *Aspergillus fumigatus* growth. Most likely, a higher MIC concentration was needed for *Aspergillus fumigatus* mold than *Candida albicans* due to the presence of dormant fungal spores. Fungal spores have thick coats that make them generally resistant to most antimicrobial agents [20]. The existence of spores in the test inoculum would also help to explain why both b-AgNPs (water ×2) and b-AgNPs (8M urea ×1) were ineffective at an MIC of 10 μg/mL. The lower MIC value of each of the 3 b-AgNPs was sufficient to inhibit *Candida albicans* yeast since it is unable to form spores. The requirement for a higher b-AgNPs MIC concentration may have also been due to the fact that fungi have different cell walls than bacteria [21]. Collectively, both the bacterial and fungal MIC results serve to

illustrate the superior action of b-AgNPs (8M urea ×2) along with their potential to act as a broad-spectrum antimicrobial agent.

The increased antimicrobial activity of the b-AgNPs (8M urea ×2) is thought to be related to, at least in part, urea's chemical action. Urea is a protein-denaturing agent that increases the solubility of hydrophobic molecules [22]. It is capable of removing affixed insoluble molecules such as fatty acids and lipids from b-AgNPs treated once with 8M urea. This would expose more soluble functional groups, such as carboxyl (-COOH), amine (-NH2), amide (-NR2) and hydroxyls (-OH), thus making the 8M urea-treated b-AgNPs more soluble and more stable while in an aqueous solution. A second 8M urea wash would have acted to remove additional insoluble groups, allowing more soluble molecules present in the cell lysate to attach to its surface. Interactions occurring between soluble b-AgNPs surfaces and cell lysate molecules would have been greatly influenced by these forces while suspended in an aqueous environment [23]. High-speed centrifugation during b-AgNPs processing would have reduced the distance between both entities, thereby increasing the potential number of interactions. In theory, a greater amount of functionalization with available soluble molecules occurred in the b-AgNPs (8M urea ×2) group receiving two separate 8M urea treatments, thereby enhancing its antimicrobial activity to a much greater degree than the b-AgNPs (8M urea ×1) receiving only a single 8M urea treatment (Figure 7).

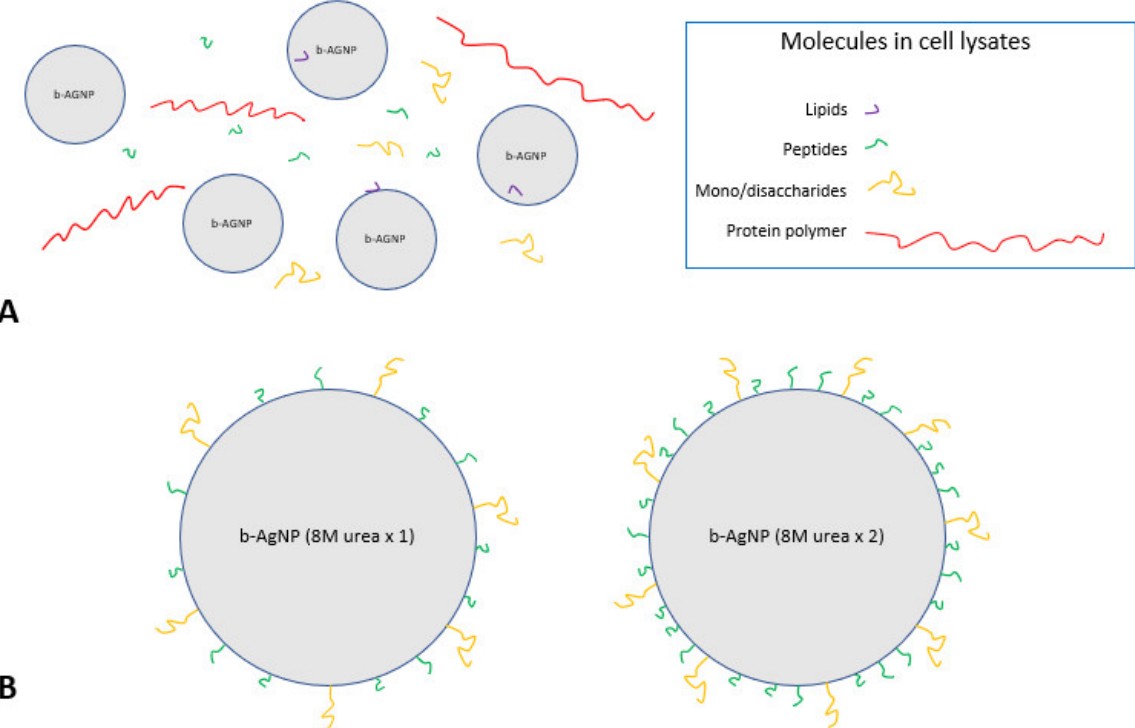

**Figure 7.** Functionalized Biogenic AgNPs. Theoretical schematic depicting how b-AgNPs treated twice with 8M urea become more thickly coated with small molecules during processing. Panel (**A**) shows b-AgNPs in close proximity to Table 1's different bacterial molecules in whole-cell lysate debris. Panel (**B**) shows greater functionalization of b-AgNPs (8M urea ×2) with a larger number of soluble molecules (nanoparticles enlarged).

Organic molecules of different composition, size, and complexity can functionalize nanoparticle surfaces. These biomolecules can range from small molecules such as lipids, vitamins, peptides, sugars to much larger polymers including proteins, enzymes, and nucleic acids. Functionalizing nanoparticle surfaces with biomolecules changes their surface composition and overall structure but leaves their bulk properties intact [24]. Molecule attachment can occur via a variety of chemical groups, including carboxylic acid, primary amine, alcohol, phosphate, and thiols. Theoretically speaking, any number of molecules

can be attached to the nanoparticle's surface. In this respect, biofunctionalized AgNPs show great potential to serve as delivery vehicles, including transport of chemotherapeutic agents and genes [25]. Additionally, biofunctionalized AgNPs coated with either chitosan or bovine serum albumin (BSA) have been shown to be effective in controlling *Streptococcus mutans*, a bacterium associated with plaque biofilms causing dental caries [26]. Chitosan-coated AgNPs are also effective against gram-negative bacteria, exhibit low cytotoxicity, and may be useful in sustained drug release, such as the antifungal agent itraconazole [27–29].

The proposed mechanisms for the increased antimicrobial activity of the b-AgNPs (8M urea ×2) are further related to the determined particle size and zeta potential. Regarding size, smaller metallic nanoparticles tend to interact more with microbe plasma membranes leading to their destruction [30]. In this regard, AgNPs ≤100 nm in diameter are used in a wide range of applications, including antimicrobial agent coatings for biomedical device, drug-delivery vehicles, imaging probes, diagnostic devices, and optoelectronic components. This is mainly due to their well-known antimicrobial activity and exceptional electrical properties [1]. Furthermore, nanoparticles must also be small enough to pass through the microbe's outer cell wall before contacting the underlying plasma membrane. It is highly unlikely that the outer cell wall of tested microbes prevented any AgNPs from reaching the underlying plasma membrane. It cannot be readily explained why the Alfa Aesar c-AgNPs measuring ~129 nm did not demonstrate antimicrobial activity while the somewhat larger b-AgNPs (8M urea ×2) measuring ~156 nm did. This suggests two distinctive points. First, that functionalized b-AgNPs >100 nm in size can also exhibit considerable antimicrobial activity. Second, that AgNPs' size alone was not a major determining factor of antimicrobial activity in the current study, as witnessed by a lack of Alfa Aesar c-AgNPs activity against test microbes. The measured −44.4 mV zeta potential for the b-AgNPs (8M urea ×2) indicates good particle stability even with the presence of some small aggregates while suspended in water.

Finally, retesting of the refrigerated AgNPs indicated that each of the three b-AgNPs had lost very little potency after a year in storage in amber microtubes protected from light (Table 6). A greater loss in potency was noted for both the b-AgNPs (water ×2) and b-AgNPs (8M urea ×1) where at least a 4× decrease in their effectiveness was noted. However, the b-AgNPs' (8M urea ×2) ability to control MSSA growth was minimally affected, changing from an MIC of 0.62 µg/mL to 1.25 µg/mL, effectively a doubling of its concentration. This finding suggests that 2 mM sodium citrate is a good choice for enhancing the long-term stability of stored AgNPs.

**Table 6.** Minimum inhibitory concentrations (MIC) of tested nanoparticles against *Staphylococcus aureus* (MSSA) one year later.

| Nanoparticle | MIC (µg/mL) | | |
|---|---|---|---|
| | *Staphylococcus aureus* (MSSA) 1st Assay ($7.5 \times 10^5$) | *Staphylococcus aureus* (MSSA) 2nd Assay ($3.0 \times 10^5$) | Difference 1st vs. 2nd Assay (±Dilutions) |
| b-AgNPs (water ×2) | 2.50 | ≥10 | ≥+2 |
| b-AgNPs (8M urea ×1) | 1.25 | 5.0 | +2 |
| b-AgNPs (8M urea ×2) | 0.62 | 1.25 | +1 |
| c-AgNPs (Alfa Aesar) | ≥10.0 | ≥10 | N/A |
| c-AgNPs (SkySpring) | ND | ≥10 | N/A |

( ) = Colony-forming units per $mL^{-1}$, ND = Not done.

## 5. Conclusions

Post-production processing of extracellular synthesized b-AgNPs by three different treatments resulted in varying solubilities that enabled a coating of their surfaces with

lysate biomolecules. Functionalization of b-AgNPs occurred as small molecules present in bacterial cell lysates became associated with their surfaces. The greatest degree of functionalization occurred in the b-AgNPs treated twice with 8M urea due to an increased solubility, which in turn imparted a greater degree of antimicrobial activity. If true, this suggests that treating b-AgNPs with two consecutive 8M urea treatments may encourage greater solubility and stability while in an aqueous solution, which results in better growth control of infectious microbes, at least under the specified test conditions. Moreover, a greater amount of functionalization in the b-AgNPs (8M urea ×2) may have also contributed to its greater stability over time. An important observation was that all three b-AgNPs demonstrated greater antibacterial capability than the c-AgNPs acquired from two commercial sources. In all cases, the two tested c-AgNPs were incapable of inhibiting the microbial growth of tested bacteria and fungi at the highest concentration tested. In summary, the lower effective concentration, broad-spectrum antimicrobial activity, small size, and long storage life of b-AgNPs (8M urea ×2) make them excellent candidates to use in a variety of applications designed to prevent microbe growth in lieu of tested c-AgNPs. Further benefits of the AgNPs (8M urea ×2) include lower required amounts for microbe control, decreased risk of human cytotoxicity, and reduction of chemical synthesis waste that eventually finds its way into the environment.

**Supplementary Materials:** The following supporting information can be downloaded at: https://www.mdpi.com/article/10.3390/applnano3040014/s1, Figure S1: The individual spectra and Voight residual fit.

**Author Contributions:** Conceptualization, T.R. and Q.Y.; Methodology, T.R., Q.Y. and M.H.; Validation, T.R., Q.Y. and M.H.; Formal Analysis, T.R. and Q.Y.; Investigation, T.R., Q.Y. and M.H.; Resources, T.R. and Q.Y.; Writing—T.R., and Q.Y.; Writing—Review & Editing, T.R., Q.Y. and M.H.; Visualization, T.R. and Q.Y. Supervision, T.R. and Q.Y; Project Administration, T.R. Funding Acquisition, T.R. All authors have read and agreed to the published version of the manuscript.

**Funding:** This research was supported in large part by a CORS grant from the Pat Capps Covey College of Allied Health Professions, University of South Alabama. This research was funded by Department of Defense/Army Research Office (DoD/ARO), grant number W911NF-18-1-0444.

**Data Availability Statement:** Data is available upon request.

**Acknowledgments:** We thank Edward Durante from the Department of Chemistry, University of South Alabama for performing both Vis spectroscopy and FT-IR analysis, as well as Alexandra Stenson, Department of Chemistry, University of South Alabama for the use of her lyophilizing equipment. We greatly thank Silas Leavesley, Department of Chemical and Biomolecular Engineering, University of South Alabama for producing the Voigt fit information. Finally, the authors would like to thank Yong Ding at George Institute of Technology for his help in performing the EDS examination.

**Conflicts of Interest:** The authors declare no conflict of interest or personal relationships that could have appeared to influence the work reported in this paper.

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
