# Peer review of "Biogenic Silver Nanoparticles Processed Twice Using 8M Urea Exhibit Superior Antibacterial and Antifungal Activity to Commercial Chemically Synthesized Counterparts"

_2673-3501, doi:10.3390/applnano3040014_

Round 1

Reviewer 1 Report

The authors in their manuscript “Biogenic Silver Nanoparticles Processed Twice Using 8M Urea exhibit superior antibacterial and antifungal activity to commercial chemically synthesized counterparts”, have synthesized silver nanoparticles using cell lysate and investigated different conditions in post treatment of the nanoparticles.

Although the authors do saw that different treatment have an effect on the antibacterial and antifungal activity compared to commercially AgNPs, my recommendation is to Reconsider after Major Revisions.

Major Revisions

1)      The authors during synthesis have only used 1mM of AgNO3. Have they tried different concentrations? Why the have chosen only 1mM?

2)      The authors have tried with Urea only the b-AgNPs. Why they haven’t applied the same procedure to the commercially available c-AgNPs. Is it possible after the Urea treatment to have different properties and the c-AgNPs?

3)      Have the authors tried to make different size nanoparticles using their method?

4)      Is there a reference of 420 nm λ max of elemental silver?

5)      Figure 2 is not conclusive as it is. Please normalise all spectra so that the individual peaks can be seen, or make two different figures

Minor Revisions

1)      L51 the term “silver substrate” is not relevant to the context of nanoparticle formation in chemical synthesis.   

2)      L95 please correct 4500x g into 4500 G

3)      L194 please correct 3,000g into 3000 G

4)      L236 what is the model name of the DLS instrument as the sentence appears incomplete “ … a (Malvern, UK)

5)      In Figure 2, please use a legend within the figure (not just A, B, e.t.c.)

Author Response

1)      The authors during synthesis have only used 1mM of AgNO3. Have they tried different concentrations? Why the have chosen only 1mM?

The authors thank the reviewer for raising this critical question. We chose this concentration based on several manuscripts that used this concentration for biological synthesis of silver nanoparticles [1-5]. In these studies, and their cited references, at 1mM AgNO3, biologically produced AgNPs showed desired morphology and a size within 100 nm, as well as good antimicrobial activity. In addition, some of these studies tested the impact of AgNO3 concentration on AgNPs production [3- 5].  These studies suggested that increase in AgNO3 concentration could increase AgNPs yield using either chemical or biological method. At 0.5mM to 3mM concentration of AgNO3, they produced AgNPs with good yield and desired size without compromising the colloidal status and uniformity of AgNPs. But at higher concentration, nanoparticle aggregation and size variation may increase [4].

The biosynthesized AgNPs used in this study exhibited longer shelf-life, suitable size, and greater antimicrobial activity compared to chemically synthesized commercial ones. At this point, the need to explore whether, or how, changes in AgNO3 concentration affect the physical and chemical properties of AgNPs is not urgent.  However, the reviewer’s suggestion of using a different concentration other than 1mM AgNO3 has value.  We now plan of evaluating the cost-effectiveness of increasing the AgNO3 concentration to find out the optimal concentration that gives rise to higher yields and even better antimicrobial action, in producing AgNPs destined for commercial applications.

2)      The authors have tried with Urea only the b-AgNPs. Why they haven’t applied the same procedure to the commercially available c-AgNPs. Is it possible after the Urea treatment to have different properties and the c-AgNPs?

We thank the reviewer for this interesting question. The 8M urea treatment was used to remove residual bacterial cell lysate material from b-AgNP samples. A similar urea treatment of the commercial c-AgNPs was not performed for two reasons. First, the Alfa Aesar AgNPs were received already suspended in 2mM sodium citrate. Second, the SkySpring AgNPs were received in a purified (99.95%) powered form. There was no need to perform a similar 8M urea treatment to “clean up” this purified sample. However, we will keep in mind that a treatment of the SkySprings commercial AgNPs with 8M urea would offer another control group for future studies.

3)      Have the authors tried to make different size nanoparticles using their method?

We appreciate the reviewer asking this question. The short answer is no, we did not. Review of preliminary batches of b-AgNPs using SEM photomicrographs showed that the nanoparticles had both the desired morphology and size needed for evaluation as an antimicrobial agent. We will continue to refine the current method in hopes of producing even smaller b-AgNPs with enhanced surface area.

4)      Is there a reference of 420 nm λ max of elemental silver?

The authors are glad that the reviewer asked this question. The maximum absorbance for elemental silver is 270 nm. We have corrected this statement to indicate that 420 nm represents “a value consistent with silver nanoparticles”. The 2014 paper by Iravani , et al is now referenced to support 420 nm. This value is further supported later on by Ejbarah, who stated AgNPs have absorption peaks between 420-480 nm. Differences in AgNPs particle size appears to be the main reason for this range.

5)      Figure 2 is not conclusive as it is. Please normalise all spectra so that the individual peaks can be seen, or make two different figures

The Chemist who ran the spectra seen in Figure 2 left the university. As a consequence, we do not have access to any file data or software needed to construct the requested normalized spectrum. Though, we do have an individual spectrum of each AgNPs but with lower image quality. These images could be submitted as Supplemental Materials.

Minor Revisions

1)      L51 the term “silver substrate” is not relevant to the context of nanoparticle formation in chemical synthesis.  

The term “silver substrate” has been removed. The sentence has been rewritten to indicate that a strong reducing agent is used to “convert a chemical compound like AgNO3 into a metallic particle”

2)      L95 please correct 4500x g into 4500 G

Corrected.

3)      L194 please correct 3,000g into 3000 G

Corrected.

4)      L236 what is the model name of the DLS instrument as the sentence appears incomplete “ … a (Malvern, UK)

Corrected. A Model ZEN 1600 Zetasizer has been included to complete this sentence.

5)      In Figure 2, please use a legend within the figure (not just A, B, e.t.c.)

Corrected.

References for item 11):

  1. Siddiqi, K.S., Husen, A. & Rao, R.A.K. A review on biosynthesis of silver nanoparticles and their biocidal properties. J Nanobiotechnol16, 14 (2018). https://doi.org/10.1186/s12951-018-0334-5
  2. Krishna Gudikandula & Singara Charya Maringanti (2016) Synthesis of silver nanoparticles by chemical and biological methods and their antimicrobial properties, Journal of Experimental Nanoscience, 11:9, 714-721, DOI: 10.1080/17458080.2016.1139196
  3. Matin Azizi, Sajjad Sedaghat, Kambiz Tahvildari, Pirouz Derakhshi & Ahad Ghaemi (2017) Synthesis of silver nanoparticles using Peganumharmala extract as a green route, Green Chemistry Letters and Reviews, 10:4, 420-427, DOI: 10.1080/17518253.2017.1395081
  4. Hamouda RA, Hussein MH, Abo-Elmagd RA, Bawazir SS. Synthesis and biological characterization of silver nanoparticles derived from the cyanobacterium Oscillatoria limnetica. Sci Rep. 2019 Sep 10;9(1):13071. doi: 10.1038/s41598-019-49444-y. PMID: 31506473; PMCID: PMC6736842.
  5. Rose, G., Soni, R., Rishi, P. & Soni, S. (2019). Optimization of the biological synthesis of silver nanoparticles using Penicillium oxalicum GRS-1 and their antimicrobial effects against common food-borne pathogens. Green Processing and Synthesis, 8(1), 144-156. https://doi.org/10.1515/gps-2018-0042

Reviewer 2 Report

 Comments to the Authors

In this manuscript the authors’ synthesized biogenic Silver Nanoparticles and demonstrated its activity towards as an antibacterial and antifungal. This research has value for the researchers in the related areas. However, the paper needs improvement before acceptance for publication. My detailed comments are as follow:

1. Authors should describe about the general properties of silver nanoparticles for this authors should use the following articles as reference:

doi.org/10.1002/slct.201900470

2. Authors should provide a histogram of silver nanoparticles.

3. The quality of Figure 2 and 3 should be improved.

4.  The writing of abstract should be improved and more concise.

5.   There are few grammatical errors. It should be improved.

Author Response

  1. Authors should describe about the general properties of silver nanoparticles for this authors should use the following articles as reference: doi.org/10.1002/slct.201900470

The authors reviewed the referenced paper, which describes the synthesis of an Ag-PF (pyrogallol-formaldehyde) resin nanocomposite. We are not sure if this was the intended article since it is very specific for this nanocomposite type.

We appreciate the reviewer’s suggestion. However, there is so much heterogeneity (size, shape, synthesis, etc.) in AgNPs literature that we would find it difficult to collectively describe their attributes in general terms. For this reason, we choose to approach AgNPs from an application perspective (e.g., antimicrobial agent) seen in the Introduction section. We feel that this approach has merit since it is found in several AgNPs reviews.

  1. Authors should provide a histogram of silver nanoparticles.

The authors would like to thank the reviewer for their suggestion. It proved invaluable in evaluating size differences seen between b-AgNPs (8M urea x2) DLS measurements using an early model Zetasizer and direct measurements using TEM photomicrographs. The histograms have been inserted in the manuscript along with average measurements and comparison of these results.

  1. The quality of Figure 2 and 3 should be improved.

Figures 2 & 3 have been enhanced (sharpened), legends have been added at the upper right corner, and captions revised.

  1.  The writing of abstract should be improved and more concise.

The abstract has been revised as suggested.

  1.   There are few grammatical errors. It should be improved.

The manuscript was checked for grammatical errors. We would appreciate the reviewer supplying comments as to where the quality of writing could be further improved.

Round 2

Reviewer 1 Report

The authors in their revised manuscript “Biogenic Silver Nanoparticles Processed Twice Using 8M Urea exhibit superior antibacterial and antifungal activity to commercial chemically synthesized counterparts”, have answered most of my previous comments.

Examining however their UV – Vis spectra on the provided supplementary material, there is a clear difference on the spectra after different treatments (water, 8M urea x2, 8M urea x1) especially in the range 520 – 680 nm. Are the observed effects only due to the nanoparticle capping layer or have the nanoparticles been modified? Have the authors tried to fit a Voigt function to the UV – spectra that would provide further evidence of the observed peaks? What was the blank used for UV – Vis spectra?

In my opinion the origin of this band and or/ peaks should be investigated further.

Therefore, my recommendation is to Reconsider after major Revision.

Minor Corrections

L58 “silver substrate” has not been changed

Author Response

- Are the observed effects only due to the nanoparticle capping layer or have the nanoparticles been modified?

This is a very interesting question. All three sets of derived nanoparticles came from the same batch of b-AgNPs and were only separated for the 3 different treatments. We therefore suspect, but cannot confirm, that the noted variation seen in these 3 nanoparticles is primarily due to a difference in their capping layers occurring during individual post-production treatments.

- Have the authors tried to fit a Voigt function to the UV – spectra that would provide further evidence of the observed peaks?

Voight fit residual figure has been included in supplemental materials. Engauge Digitizer was used to extract all the datapoints from each sample curve and then fit them to a series of Voigt curves using a MATLAB script (4-peak model).

- What was the blank used for UV – Vis spectra?

A 2mM sodium citrate solution was used as a blank since each tested nanoparticle type was suspended in this solution.  

- L58 “silver substrate” has not been changed

Corrected this version.